# Research on Insurance Method for Energetic Materials on Information Self-Destruction Chips

**DOI:** 10.3390/mi13060875

**Published:** 2022-05-31

**Authors:** Hengzhen Feng, Wenzhong Lou, Bo He, Sining Lv, Wenting Su

**Affiliations:** School of Mechatronical Engineering, Beijing Institute of Technology, Beijing 100811, China; cqrhb0928@126.com (B.H.); lvsining@bit.edu.cn (S.L.); 3120210187@bit.edu.cn (W.S.)

**Keywords:** energetic materials, information insurance, physical self-destruction, insurance actuator, physical isolation

## Abstract

Detonation waves released by energetic materials provide an important means of physical self-destruction (Psd) for information storage chips (ISCs) in the information insurance field and offer advantages that include a rapid response and low driving energy. The high electrical sensitivity of energetic materials means that they are easily triggered by leakage currents and electrostatic forces. Therefore, a Psd module based on a graphene-based insurance actuator heterogeneously integrated with energetic materials is proposed. First, the force–balance relation between the electrostatic van der Waals force and the elastic recovery force of the insurance actuator’s graphene electrode is established to realize physical isolation and an electrical interconnection between the energetic materials and the peripheral electrical systems. Second, a numerical analysis of the detonation wave stress of the energetic materials in the air domain is performed, and the copper azide dosage required to achieve reliable ISC Psd is obtained. Third, the insurance actuator is prepared via graphene thin film processing and copper azide is prepared via an in situ reaction. The experimental results show that the energetic materials proposed can release physical isolation within 14 μs and can achieve ISC Psd under the application of a voltage signal (4.4–4.65 V). Copper azide (0.45–0.52 mg) can achieve physical damage over an ISC area (23.37–35.84 mm^2^) within an assembly gap (0.05–0.25 mm) between copper azide and ISC. The proposed method has high applicability for information insurance.

## 1. Introduction

Physical self-destruction (Psd) represents an important method to improve the information insurance performance of information storage chips (ISCs) [1,2]. The German Federal Ministry of Education and Research invested more than €25 million in the ZEUS Trusted Electronics Program in 2020, which is dedicated to developing the information insurance for information storage chips [3]. Under the support of this project, the Psd module with a multi-sensor stack was developed by Fraunhofer Institute. To date, numerous research institutions have announced a variety of schemes (e.g., using high-voltage excitation, electrochemical degradation, detonation stress waves, and other methods) to achieve the Psd of ISCs. For example, the US Defense Advanced Research Projects Agency (DARPA) launched the Vanishing Programmable Resources (VAPR) project in 2013 [4]. According to the self-destruction requirements for military-grade electronic equipment that were developed by the US military, the VAPR project [5] was based on a “pressure release trigger decomposition” approach, which was intended to verify that military electronic systems had the ability to be destroyed in a physical manner; this would prevent the leakage of secret information and manufacturing processes after reverse engineering of the core electronic devices. In 2015, the Xerox Palo Alto Research Center released a self-destructing chip [6]. The mechanism was triggered by a laser or wireless signal, causing the chip to heat up and crack the electronic device by heat accumulation. After the thermal shock occurred, the substrate broke within a few seconds, and the device could then not be restored. However, these methods relied on the application of external high-power energy, e.g., laser signals, and thus, apply to a limited number of scenarios. Therefore, energetic materials [7,8] have been used to generate detonation waves (GPa level) in microseconds via voltage excitation (below 5 V) to achieve Psd. However, because of the low driving thresholds of these energetic materials, the improvement of the anti-interference capabilities of these materials has become another core issue in this field. Many anti-interference ability protection measures use a transient voltage suppression (TVS) diode [9,10,11,12] to absorb transient energy. The voltage clamp was applied between two poles and was set at a predetermined value that could not be adjusted for use under different environments [13,14,15,16], but its energy absorption performance was limited [17,18]. In 2020, the Beijing Institute of Technology announced OFF-ON-OFF microelectromechanical systems based on a modified Paschen’s law [19]; in this system, a comb-shaped actuator (electrode gap: 1–10 µm) was able to reduce the threshold voltage to 130 V for air breakdown [19], and the electrical state was switched in an OFF-ON-OFF sequence to complete energy grooming for abnormal interference signals. In addition, a metal-based “OFF-ON” actuator was designed based on corona discharge theory to achieve air breakdown at a threshold voltage [20,21,22,23]. These schemes can be used to provide an energy grooming function to achieve insurance control of the energetic materials. However, the threshold voltage generally still exceeds 45 V [24,25], which represents a comparatively high energy burden for Psd modules. In contrast, only the “ON-OFF” actuator offers the advantage of low voltage operation. In 2010, Rossi from LAAS in France developed three actuators: an electrothermal evaporation-breaking solid actuator, an electro-explosion-breaking solid actuator, and an electrothermal welding-conducting solid actuator. These three actuator types mainly rely on the thermal accumulation of their metal electrodes. This thermal energy acts on the sensitive mechanism of the actuator [26,27], causing electrode melting/vaporization and other phenomena. The voltage pulses for these actuators are distributed over a range from 3.3 V to 8 V, and the response times of the actuator states (i.e., ON-OFF) are on the microsecond scale. In 2014, Beijing Institute of Technology presented a solid-state actuator based on a multilayer metal stack. This type of actuator can melt out completely under a threshold voltage of 1.5 V. The response time distribution for the actuator ranges from 10 ms to 150 ms [28,29]. These actuators can only isolate the energetic materials from the peripheral electrical system by parallel connections, but the isolation method is only intended to provide electrical isolation, and thus, cannot shield the materials from the effects of the leakage currents. Therefore, based on the insurance requirements for energetic materials discussed above, this paper presents an insurance actuator that provides physical isolation from peripheral electrical systems through functional integration with the energetic materials, which also reduces triggering of the materials by leakage currents. The proposed technique has a strong practical application value.

## 2. Theory

### 2.1. Design of Insurance Actuator Model for Energetic Materials

In this paper, ISC Psd is achieved using a detonation wave that is generated by energetic materials in a micro-region and propagates in the air domain. The scheme offers advantages that include small device size, low driving energy requirements, and fast response times. The energetic materials are mainly composed of a semiconductor bridge (SCB) and copper azide. The detonation wave for copper azide requires high-temperature plasma that is released by the SCB. When the external electrical signal acts on the SCB, the bridge region goes through a solid heating–liquid heating–silicon gasification–silicon vapor ionization sequence to form a high-temperature plasma. This high-temperature plasma is diffused into the copper azide by micro-convection and thus realizes the detonation energy output. However, because of the limitations of the high electrostatic sensitivity of copper azide, it can also be easily driven by external leakage currents or electrostatic, thus resulting in false triggering. Therefore, this paper presents the design of a Psd module based on matching the design of the insurance actuator and the energetic materials. If the physical isolation and connection between energetic materials and the peripheral electrical systems can be achieved, the insurance of energetic materials can be improved.

The insurance actuator consists of a three-layer graphene electrode that is stacked together with a dielectric layer and is connected in series with the SCB, as illustrated in Figure 1. The insurance actuator has “absorption” and “isolation” states. In these two states, the “absorption” state represents the interconnection of the energetic materials with the peripheral electrical systems, and the “isolation” state represents the physical isolation of the energetic materials from the peripheral electrical systems. The ISC self-destruction mechanism based on the energetic materials and the insurance actuator is described as follows and is shown in Figure 2.

The bottom graphene electrode induces V_signal-1_, an electric field that can be generated between the bottom layer graphene electrode (BLGE) and the boron nitride dielectric layer. The top layer graphene electrode (TLGE) induces an electrostatic van der Waals force to overcome the elastic recovery force and makes contact with the intermediate layer graphene electrode (ILGE). At this time, the energetic materials are connected in series with the peripheral electrical system. When the ILGE induces *V*_signal-2_, the SCB then produces a high-temperature plasma that is used to drive the copper azide. Finally, the copper azide releases the detonation wave, which is then transferred to the ISCs, allowing for the Psd to be achieved. In addition, when *V*_signal-1_ is reset, the electric field in the dielectric layer then vanishes. The TLGE resets under the action of the elastic recovery force alone, thus causing the energetic materials to be isolated physically from the peripheral electrical system. Due to the appropriate functional matching design between the actuator and the energetic materials, the energetic materials are suitable for use within the information insurance field.

To achieve TLGE absorption, it is important to build a balance relation between the electrostatic van der Waals force and the elastic recovery force. A force-balance equation is constructed as shown in Equation (1), and the elastic recovery force is described using Equation (2):(1)$$F_{\mathrm{re}}=kd=n\frac{8Ewt^{3}}{L^{3}}\left(l-z\right)$$
(2)$$F_{\mathrm{Con}}=F_{\mathrm{van}}-F_{\mathrm{re}}=\frac{A_{H}A}{12\pi z^{2}}-kd=\frac{A_{H}A}{12\pi z^{2}}-n\frac{8Ewt^{3}}{L^{3}}\left(l-z\right)$$

In these equations, *E* is Young’s modulus for graphene, *w*, *t*, and *L* are the width, thickness, and length of the TLGE, respectively, and *z* is the gap between the TLGE and the ILGE. Additionally, *A_H_* is the Hamaker constant of graphene, which is taken to be 4.7 × 10^−19^ J, *A* is the contact area, and *K* is the spring constant of the graphene beam.

To increase the TLGE motion displacement, this paper proposes four electrode models in combination with the mechanical relation given in Equation (1). The relationship between the electrode length and the motion displacement was obtained as shown in Figure 3, and a mechanical simulation analysis of the TLGE was performed as shown in Figure 4.

According to Figure 3, the four models show a trend that when the gap between the TLGE and the ILGE increases, the length of the TLGE initially increases and then decreases. When this gap is 4 nm, the length of the D model is the longest at 22.96 μm, which is longer than all the other models.

In this paper, the von mises and displacement numerical analyses of four models are carried out under the gap (4 nm) between the TLGE and the ILGE in Figure 4.

When the TLGE induced the electrostatic Van der Waals force for 1 µs, the A model produces the maximum von mises and motion displacement (10.4 MPa, 0.14 μm), the C model produces the minimum von mises and motion displacement (1.44 MPa, 0.08 μm), in which the maximum von mises is less than the graphene fracture strength, and the minimum displacement is greater than 4 nm. It can be seen that the four TLGE models of the insurance actuator can satisfy the transformation from the “absorption” to “isolation” states under steady-state conditions. The relationships among the “absorption” time, the electrode model, and the displacement are analyzed, with the results illustrated in Figure 5.

According to Figure 5, under the same displacement conditions, the graphene electrode (D model) shows the shortest motion response time of 4.47 µs. The absorption time with the A model is 5.5 µs. In addition, when using the D model as an example, the absorption time of the TLGE consists of three stages: the pre-suction stage, the liner suction stage, and the critical suction stage.

When the electrode induces an electrostatic van der Waals force and the displacement is within the 0–0.5 nm range, the absorption time is close to 1.5 µs, and the velocity of the movement is 0.25 × 10^−3^ m/s. When the displacement is in the range from 0.5 nm to 5.3 nm, the absorption time shows a linear trend, and the maximum velocity reaches 3.2 × 10^−3^ m/s. Subsequently, when the TLGE moves toward the ILGE, the TLGE velocity falls to 0.47 × 10^−3^ m/s. In other words, the TLGE’s motion is constrained strongly by the electrostatic field and the restoring force during the initial and final stages, respectively.

According to Figure 5b, the resultant force acting on the TLGE increases before weakening during the absorption state, and the complete movement process can be divided into two distinct sections. Using the D model as an example, during the period from 0 to2.5 µs, the resultant force increases, thus causing the displacement gradient to increase. Then, during the period from 2.5 to 4.5 µs, the displacement gradient decreases, with the resultant force also decreasing. The resultant force in the D model is greater than that in the other electrode models, where the maximum absorption time follows the relation t_A_ > t_B_ > t_C_ > t_D_. In summary, the D model is used for the TLGE. The electrode thickness was calculated to be 0.68 nm, and the contact area for the top electrode layer is 1.69 µm^2^.

### 2.2. Design of Energy Transfer Model for Energetic Materials

The Psd module proposed in this paper mainly includes the charge mechanism, the SCB, copper azide, and the ISCs. Copper azide represents the core self-destruction element and relies on an in situ reaction. The transmission of the detonation stress wave is performed based on a high-speed combustion material model. The dimensions of the copper azide element are Ø1 × 0.5 mm^2^, and the charge dose is in the 0.45–0.52 mg range. The stress wave of the copper azide energy transfer process over time is shown in Figure 6. A simplified model of the assembly relationship between the copper azide and the ISC is shown in Figure 6a [30].

The assembly gap between these two elements is used to enhance the growth of the detonation wave in the air domain such that it strikes the surface of the ISC with sufficiently high intensity to achieve ISC Psd. In this paper, the transmission analysis of the detonation waves in the air domain is achieved with the aid of LS-DYNA software, with results as shown in Figure 6b.

According to the results in Figure 6b, the air gap between the copper azide in the energetic material and the ISC can be divided into 17 nodes (A–R nodes), where the gap between adjacent nodes is 0.05 mm. The stress versus time relationship can be obtained from the LS-DYNA analysis: the stress at each node in the air domain increases initially and then decreases, and the stress wave attenuates rapidly beyond the backward position of the node.

In combination with the structural critical strength (1 Gpa) of the plastic chip, when the assembly gap is 0.05 mm (A node), the maximum stress in the air domain reaches 3.15 GPa, and its duration is 25 µs; however, when the assembly gap is 0.35 mm (G node), the maximum stress in the air domain reaches 1.25 GPa, and the duration is 11 µs. In this paper, the A–G node is selected as the upper limit of the air gap between the copper azide and the ISC assembly.

Figure 6c shows the stress attenuation relationship of the detonation wave at each node in the air domain. When the node is located within the A–G range, the attenuation rate of the detonation wave reaches 4.2–10.4 GPa/mm. When the node is located within the H–R range, the stress wave attenuation rate then reaches 0.7–2.6 GPa/m. In other words, because of the serious attenuation of the stress during the initial stage (from the A node), the stress value at the H–R node is not sufficient to achieve ISC Psd. Therefore, the assembly gap between the copper azide and the ISC must be in the 0.05–0.25 mm range.

## 3. Experimental

As shown in Figure 7, the fabrication process flow includes two modules.


**MI. Preparation process for the insurance actuator.**


Step I. Graphene (thickness: 0.46 nm) is mechanically exfoliated from highly-directional pyrolyzed graphite onto silicon wafers that have been thermally oxidized to have a SiO_2_ film (40 nm) coating. The BLGE is then patterned via ion beam lithography to form the electrostatic induction unit of the insurance actuator.

Step II. A boron nitride layer (1.3 nm) is sputtered onto the BLGE, and the electrostatic field induction region is realized via lithographic processing.

Step III. A graphene (0.46 nm) electrode is prepared via a similar process (see Step I) on top of the boron nitride layer, forming the ILGE, which is then patterned by ion beam lithography.

Step IV. A SiO_2_ mask is formed by plasma-enhanced chemical vapor deposition (CVD) on the ILGE; this layer then forms the gap between the TLGE and the ILGE.

Step V. The TLGE (0.46 nm) is mechanically striped on the top of the SiO_2_ layer via ion beam lithography, and the SiO_2_ mask is etched using a buffered HF solution to release the gap between the ILGE and the TLGE. Finally, the BLGE, the ILGE, and the TLGE pad, which connect in series with the peripheral electrical system, are formed by evaporation of Pt onto the actuator surface.


**MII. Preparation process for the copper azide.**


Step I. First, polystyrene (PS) microspheres are selected as templates; microspheres (PS/Cu) are prepared by electroless Cu plating on the PS microsphere surfaces. Second, these PS/Cu microspheres are preformed and sintered at 400 °C in N_2_ and air, respectively. Finally, hollow Cu microspheres (thickness: 100 nm) and hollow CuO microspheres (thickness: 200 nm) are prepared.

Step II. The Cu and CuO microspheres are then pressed into a charging mechanism to form a nanoporous copper layer.

Step III. A nanoporous copper chip is then formed by pre-pressing of a cartridge ring.

Step IV. An in situ reaction with HN_3_ is performed at 55 °C for 24 h to create the copper azide.

The process above thus verified that copper azide can be prepared successfully via the proposed in situ reaction.

The insensitive SCB, which has a strong overcurrent ability, was selected in this study. This SCB can improve the insurance of copper azide, and the resistance in the bridge area was between 1.1–1.2 Ω.

## 4. Test and Measurement

According to the mechanism illustrated in Figure 2, the absorption behavior in the insurance actuator test was measured under various driving voltages (*V_signal-1_* = 1.30–2.10 V). In addition, multiple absorption fatigue test cycles were performed at the same driving voltage, and the relationship between the TLGE absorption time and the number of test cycles was obtained, as shown in Figure 8.

Over the same number of test cycles (cycles = 1–14), the driving voltage (*V*_signal-1_) of the BLGE gradient ranges from 1.30 V to 2.10 V, and the absorption time of the TLGE decreases, with a distribution within the 10–14 µs range. Under the same driving voltage, the absorption time increases with increasing numbers of cycles. When the number of cycles reaches 20, the absorption time then exceeds 35 µs.

Based on the experimental conclusions drawn above, the dosage of copper azide is 0.52 mg; the Psd test results obtained for the ISCs are shown in Figure 9. When the number of cycles = 5, *V*_signal-1_ = 1.728 V, and *V*_signal-2_ = 4.56 V, the Psd time (from TLGE absorption to detonation wave generation) is 10 µs. When the number of cycles = 15, *V*_signal-1_ = 2.1 V, and *V*_signal-2_ = 4.48 V, the Psd time is 14 µs. In summary, when the number of cycles ranged from 1 to 14, the variation in the Psd response time was only 40%. When the number of cycles reached 20, the Psd time reached 32 µs, which is 2.2 times longer than the original Psd time, and the mechanical response of the TLGE showed fatigue accumulation behavior.

In addition, the leakage currents between the bottom and interlayer electrodes were measured in both the absorption and isolation states, and the experimental results are shown in Figure 9d.

As Figure 9e shows, when the actuator is in the absorption state, the leakage current (*I_B−I_*), which is distributed from 0.068 to 0.073 µA within the V_signal-2_ range from 0 to 0.9 V, decreases. The main reason for this behavior is that the driving voltage of the BLGE (V_signal-1_) is higher than that of the ILGE (V_signal-2_), and their currents are in opposite directions. The leakage current is offset with increasing V_signal-2_, and when V_signal-2_ exceeds 0.9 V, the leakage current increases linearly with increasing voltage, with the linear rate of increase reaching 0.062 µA/V. When the TLGE is in the isolation state, the leakage current increases linearly with increasing V_signal-2_, and the linear rate of increase reaches 0.053 µA/V.

In addition, the Psd area of the ISC was calibrated for 10 groups, with results as shown in Table 1.

In summary, copper azide (0.52 mg) can achieve physical damage over an ISC area of 23.37–35.84 mm^2^ and the Psd of the ISC within an assembly gap range of 0.05–0.25 mm. The proposed scheme has extensive application value in information insurance.

## 5. Conclusions

In this paper, energetic materials are used to achieve ISC Psd. To prevent the energetic materials from being influenced by leakage currents, a graphene-based insurance actuator is proposed. During matching testing between the actuator and the energetic materials, it was concluded that the minimum absorption time for the TLGE is 10 µs. When the number of absorption/isolation cycles ranged from 1 to14, the variation in the Psd response time was only 40%. When the number of cycles reached 20, the Psd time reached 32 µs, which is 2.2 times longer than the original Psd time, and the mechanical response of the TLGE showed fatigue accumulation. In addition, when *V_signal-1_* was 2.10 V, *V_signal-2_* reached 4.56 V, the TLGE was in the absorption state, and ISC Psd could be achieved, with the maximum leakage current for the insurance actuator reaching 0.302 µA. Additionally, when the TLGE was isolated, the maximum leakage current was 0.26 µA, and the energetic materials were isolated from all peripheral electrical system effects completely.

## Figures and Tables

**Figure 1 micromachines-13-00875-f001:**
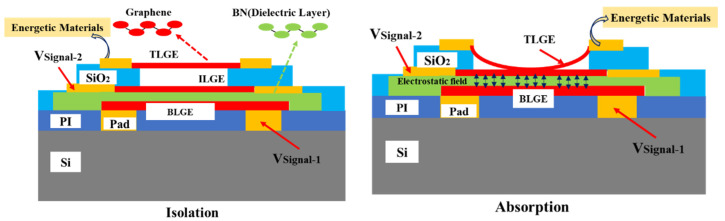
Functional matching model of insurance actuator and energetic materials.

**Figure 2 micromachines-13-00875-f002:**
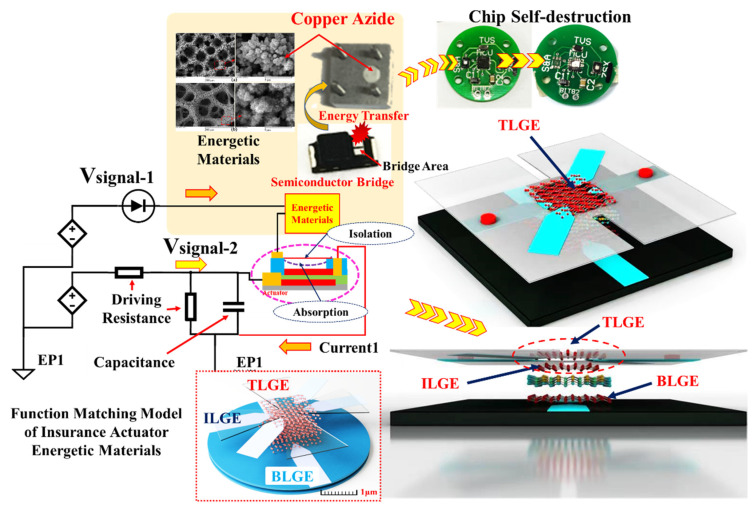
ISC self-destruction mechanism based on energetic materials and insurance actuator.

**Figure 3 micromachines-13-00875-f003:**
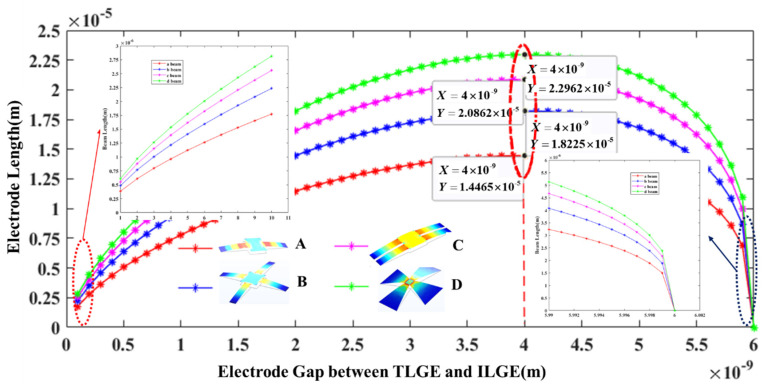
Relationship between the electrode length and the motion displacement.

**Figure 4 micromachines-13-00875-f004:**
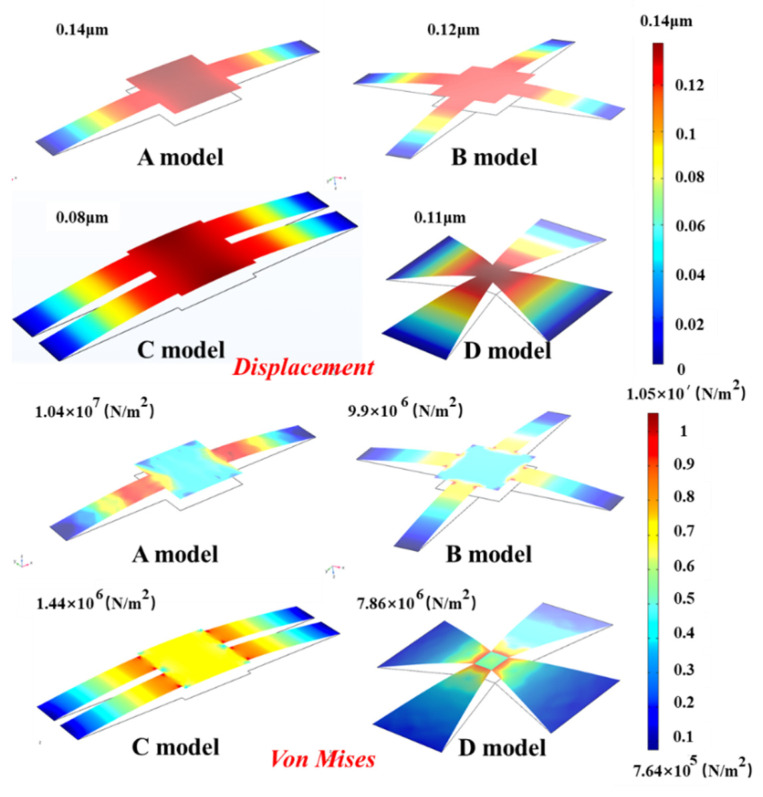
Mechanical simulation analysis of the TLGE.

**Figure 5 micromachines-13-00875-f005:**
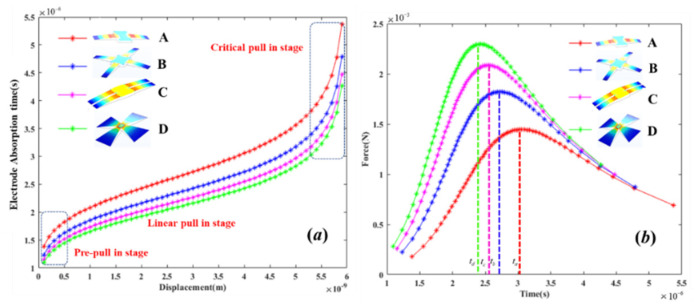
Relationships among absorption time, electrode model, and displacement. (**a**) Relationship between absorption time and displacement for four models. (**b**) Relationship between Force and absorption time for four models.

**Figure 6 micromachines-13-00875-f006:**
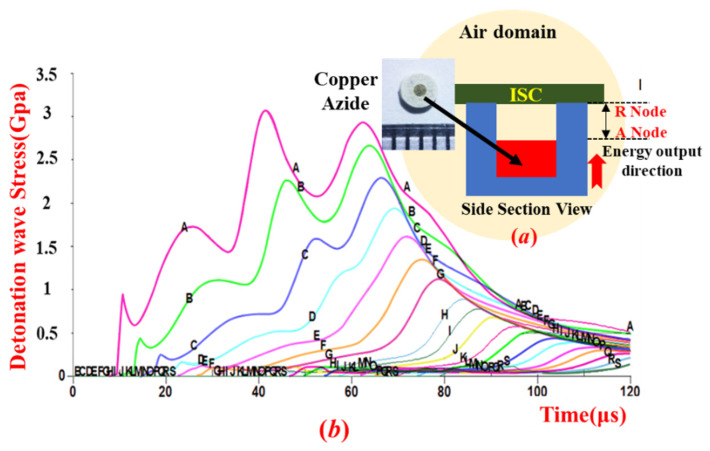
Stress wave of copper azide energy transfer over time. (**a**) The model of Psd module. (**b**)detonation wave stress variation curve by time in air domain. (**c**) The relationship between detonation wave stress, attenuation rate under different air gap.

**Figure 7 micromachines-13-00875-f007:**
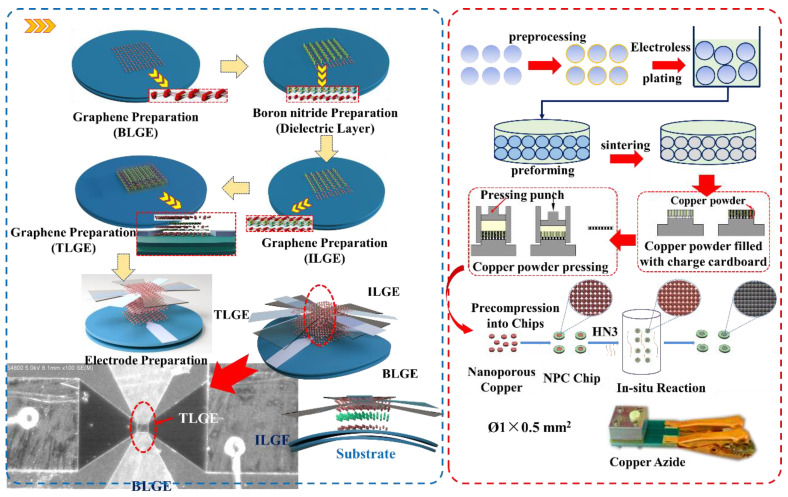
Fabrication processes of insurance actuator and copper azide.

**Figure 8 micromachines-13-00875-f008:**
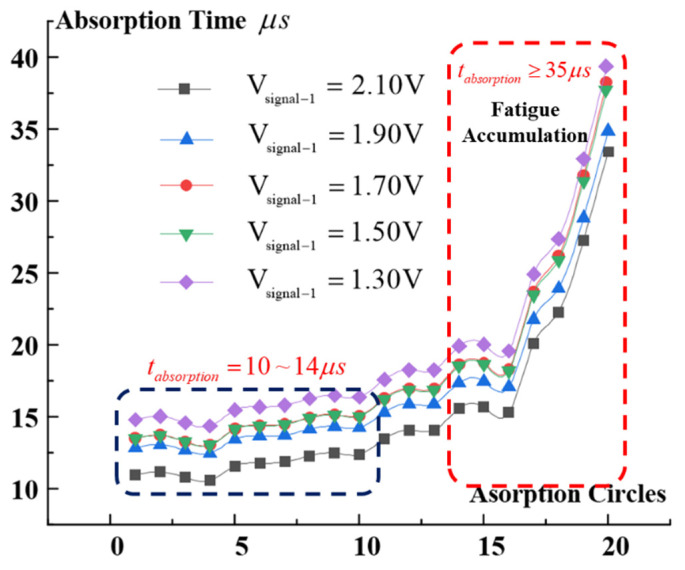
Absorption time changes with driving voltage and number of test cycles.

**Figure 9 micromachines-13-00875-f009:**
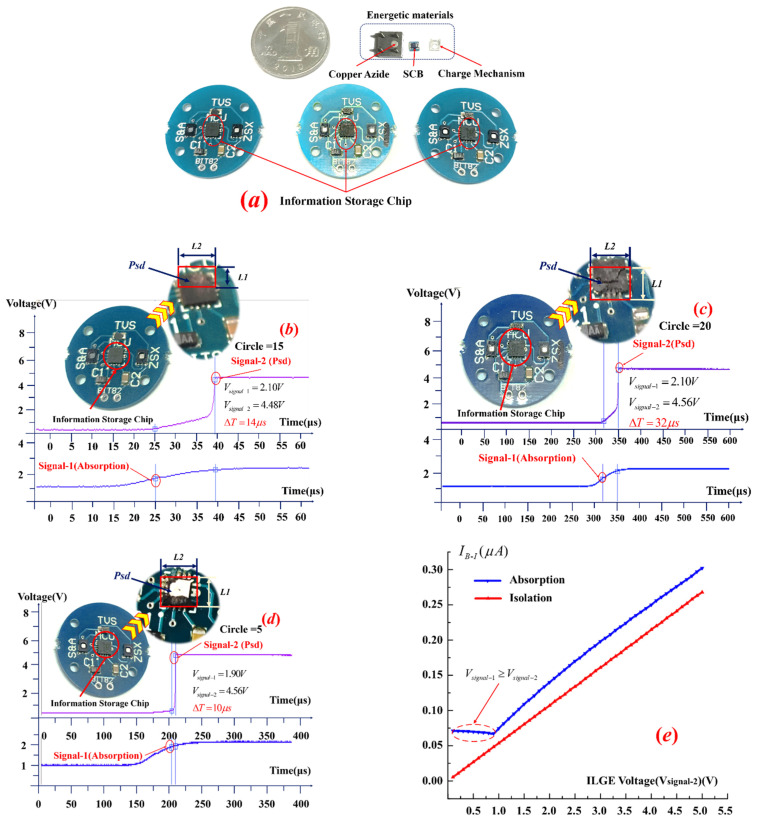
(**a**) ISC Psd module, (**b**–**d**) Psd times for different numbers of test cycles, and (**e**) leakage currents between the BLGE and the ILGE under the absorption/isolation states.

**Table 1 micromachines-13-00875-t001:** The Psd area of the ISC.

	1	2	3	4	5	6	7	8	9	10
*L*_1_(mm)	6.3	5.8	4.7	6.4	6.9	4.8	5.6	6.7	4.5	5.7
*L*_2_(mm)	4.5	5.9	6.4	4.6	4.8	6.1	6.4	3.7	5.9	4.1

## Data Availability

All other data are available from the corresponding authors upon reasonable request.

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
