# Peer review of "Research on Insurance Method for Energetic Materials on Information Self-Destruction Chips"

_micromachines, 2022, doi:10.3390/mi13060875_

Round 1
Reviewer 1 Report
This paper introduced a graphene based information self-destruction chip. However, many conceptions need to be redescribed clearly.
- There are many errors in the English expressions, which need to be carefully revised.
- It is hard to understand the paragraph in line 88~97. It looks like a paper template. Please check out the paper carefully before submitting.
- The cross section structure of ISC shown in Fig. 1 is not clear. Please provide the 3D model to illustrate the working principle.
- More explanations in Fig. 2 should be made. The simulation results of C model in “Displacement” and “Von Mises” are quite different from the others. In “Displacement”, the maximum displacement of A, B, D model occur in the middle plate. While in the C model, the maximum displacement occurs in the edge of the suspended beam. Why this phenomenon happens.
In “Von Mises”, the maximum stress of A, B, D model occur in the connection between the beam and the middle plate. While in the C model, the maximum stress still occurs in the edge of the suspended beam. Why this phenomenon happens.
In addition, the figures of C model shown in “Displacement” and “Von Mises” seem to be the same, which make the whole analysis to be suspicious.
- Fig.3 is confused. What is the meaning of “beam length” in the Y-axis, and what is the meaning of “electrode length” in the legend. If they are the same, please unify the word in this paper, and highlight the structure in Fig.2.
- Fig. 8 shown in this paper is not persuasive. Please add more SEM figures of the ISC to show the chip can be fabricated.
Author Response
I feel very privileged to receive comments and guidance from reviewers,and revise and improve for each comments, the specific amendments are in the attachment.

Reviewer 2 Report
- Introduction template materials are shown in page. 2
- The fusion between mechanical energy and electrical energy is good approach.
- it is necessary to mention the novelty of this work. For example, readers may be wonder whether transient electronics of actuator are fist work
- Entirely, authors should stress the novelty of this work. The reviewer think that this paper just lists experimental results.
Author Response

(The authors gave the same response as above.)

Reviewer 3 Report
The present paper deals with the fabrication of an Insurance Method for Self-Destruction Chips based on a copper azide reaction. This is an original study but some parts are not detailed and therefore it is difficult to understand all aspect of the concept and device. After presenting the operation of the device the authors briefly present the fabrication and characterization in terms of reaction time.
I recommend the publication after following revisions.
1st, the state of the art is not timely. Important paper such as the three below should be included.
- J. Fleck et al., « Controlled Substrate Destruction Using Nanothermite », Propellants, Explosives, Pyrotechnics, vol. 42, no 6, p. 579‑584, 2017
- Nicollet et al, « Fast circuit breaker based on integration of Al/CuO nanothermites », Sensors and Actuators A: Physical, vol. 273, p. 249‑255, 2018
- F Sevely et al. «Design, Fabrication and Testing of a Miniaturized end-of-life Ultimate Security Device Using Reactive Composites» 45th International Pyrotechnics Society Seminar 2022
2nd, the copper azide being the core element of the concept, more detail is required : mass or volume deposited. The detail of the reaction Is also requested to better understand.
3rd, there is no proof of the destruction of the electronic chip. This is required. In addition, what is the maximun size of the chip that can be destroyed using this technique.
4th, the description of the SCB that is used is also required.
Author Response

(The authors gave the same response as above.)

Round 2
Reviewer 2 Report
Thank you for reflecting comments
Reviewer 3 Report
The authors have revised and greatly improved the manuscript. Can be accepted for publication as it is.